# ThinkAfford: Enhancing VLM Reasoning for Affordance Grounding in 3D Scenes

## Abstract

Task-driven affordance grounding in 3D scenes is crucial for embodied AI agents to identify and operate functional interactive elements (*e.g.*, switches, hinges, handles) and thereby accomplish their objectives. However, current approaches have the following limitations: purely 3D point cloud pipelines struggle to generalize across scenes and categories, while 2D-driven methods guided by generic vision–language models often miss small, functionally distinct parts and produce view-dependent, inconsistent results. We introduce **ThinkAfford**, a coarse-to-fine RGB-D framework for grounding natural-language instructions to fine-grained 3D affordances in cluttered scenes. The coarse stage uses vision-language reasoning to efficiently prune thousands of frames to a compact set of relevant candidate views, leveraging context and relational cues to avoid exhaustive search. The fine stage then focuses on functional parts: it produces affordance-centric proposals that remain stable across viewpoints, and employs an instruction-guided selector fine-tuned with Group Relative Policy Optimization (GRPO) to enhance fine-grained spatial reasoning, by explicitly rewarding choices that satisfy attribute, relational, and geometric constraints. Experiments on SceneFun3D demonstrate state-of-the-art performance, achieving 14.97% AP25 on the test split—a 70.1% relative improvement over the previous SOTA method. Our results show that this structured decomposition, combined with fine-grained spatial reasoning, effectively bridges the gap between high-level language understanding and precise 3D affordance localization. The code will be made available for future exploration.

## 1 Introduction

Grounding task instructions (typically expressed by natural language) to functional object parts within complex 3D environments is a fundamental problem in embodied AI. This problem, referred to as task-driven 3D affordance grounding (Delitzas et al., 2024), is critical for enabling precise action execution and reliable human-robot collaboration. Despite recent progress, the task remains largely unsolved due to challenges such as occlusion, viewpoint variability, and the compositional nature of natural language (Wu et al., 2025; Zheng et al., 2025b). In practice, agents must distinguish small, functionally critical elements (*e.g.*, a "graspable handle" or "pushable button") from numerous distractors, where minor misalignments may lead to task failure (Li et al., 2024b).

A straightforward way to explore this problem is to develop a language-driven part segmentation pipeline based on point clouds (Takmaz et al., 2023; Li et al., 2024c; Wu et al., 2025; Chu et al., 2025). While point cloud-based affordance grounding naturally offers accurate 3D geometry and viewpoint invariance, such methods suffer from poor generalization across scenes and categories due to the scarcity of labeled 3D data. More importantly, the absence of a foundation model that aligns language with 3D geometry hampers the interpretation of complex relational instructions and further undermines generalization. These limitations motivate us to move beyond purely 3D pipelines and instead exploit 2D VLMs on RGBD sequences, leveraging their ability to parse task instructions and reason over spatial and relational cues to guide functional parts grounding.

In fact, we are not the only ones leveraging 2D VLMs to interpret RGB-D sequences for 3D scene affordance grounding. One prior work, Fun3DU (Corsetti et al., 2025), adopts a three-step pipeline: first, generating 2D candidate masks using a segmentation model; then, selecting a specific view with a VLM; and finally, back-projecting the selected 2D mask into 3D to obtain a 3D affordance

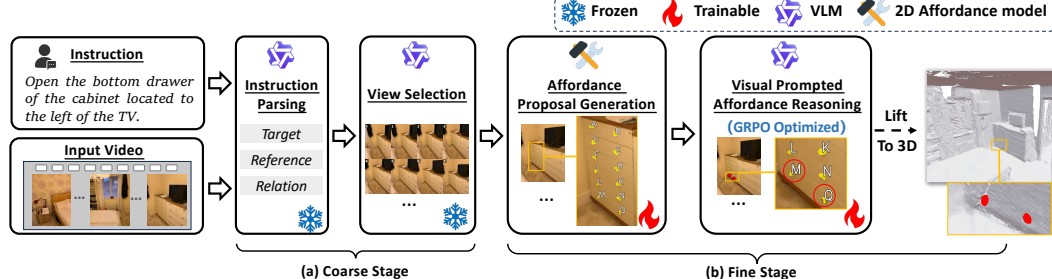

Figure 1: **An overview of ThinkAfford.** Given a natural-language instruction for operating an object in a 3D scene, ThinkAfford localizes the target affordance in two stages. In the coarse stage, a VLM parses the instruction and performs relation-aware scoring over the RGB-D sequence to prune context-irrelevant frames, retaining an instruction-relevant view set. In the fine stage, an Affordance Proposal Generation module generates viewpoint-robust proposals per image, and a GRPO-optimized Visual Prompted Affordance Reasoning module selects regions that best satisfy attribute, relational, and geometric constraints. The selected 2D evidence is then back-projected and fused into a consistent 3D affordance map.

mask. However, this view-selection-after-segmentation paradigm is brittle for small or fine-grained objects and in cluttered environments, where 2D predictors are inherently sensitive to scale and viewpoint (Zhao et al., 2024). In addition, it heavily depends on the VLM's coarse spatial understanding (Liu et al., 2023; Zheng et al., 2025a) and lacks a mechanism to accurately localize relational or attribute-specified targets (e.g., "the fifth drawer of the cabinet to the left of the TV"). Moreover, the entire pipeline operates in a purely zero-shot manner, and without training on task-specific data, the reasoning capability of VLMs cannot be fully realized.

To this end, this paper proposes **ThinkAfford**, a two-stage RGB-D framework that decomposes the grounding task into context-aware view selection and affordance-conditioned fine localization. Specifically, our method comprises two stages, as shown in Fig 1 The coarse stage parses natural-language instructions using a vision-language model (VLM) and ranks RGB-D keyframes via relation-aware scoring, pruning away irrelevant views while retaining a compact subset most likely to satisfy the spatial semantics of the instruction. The fine stage performs affordance-conditioned proposal generation and 2D-to-3D lifting. Given the selected frames, an Affordance Proposal Generation module produces target-level affordance masks, which are then filtered using the VLM to identify proposals whose fine-grained regions best match the instructed attributes and spatial relationships. The selected evidence is then back-projected and fused in 3D, yielding a consistent and localized affordance prediction.

While this two-stage pipeline enables effective task completion, standard affordance models often fall short in cluttered scenes, where objects may be off-center, heavily occluded, or small. To address this, we introduce the **Affordance Proposal Generation (APG)** module, a compact model trained within our framework to decouple affordance prediction from object identity. APG is designed to produce viewpoint-robust, fine-grained masks that generalize across diverse scene complexities. To further improve spatial reasoning and small-object sensitivity, a **Visual Prompted Affordance Reasoning (VPAR)** is introduced to perform visual-prompted affordance reasoning. VPAR is trained using Group Relative Policy Optimization (GRPO) Shao et al. (2024) in a one-of-many selection setting, enabling it to identify the candidate region that best satisfies the compositional and relational constraints embedded in natural-language instructions. In summary, ThinkAfford is a two-stage RGB-D framework that performs VLM-guided view pruning, followed by affordance-conditioned proposal generation and RL-based region selection, culminating in 2D-to-3D lifting. This design enables fine-grained 3D affordance localization under complex, open-ended instructions in large, cluttered environments.

Extensive experiments on SceneFun3D corroborate the effectiveness of the proposed framework. On the test split, ThinkAfford surpasses the strong baseline Fun3DU by +70.1% relative in AP25, respectively. On the validation split, it achieves 22.69% AP25, corresponding to +29.7% relative gains over prior methods. Ablation studies further demonstrate the importance of the proposed

components—particularly the Affordance Proposal Generation (APG) and the GRPO-trained VLM (VPAR). This demonstrates that our structured decomposition, combined with affordance conditioning and grounded selection, effectively resolves the granularity and fidelity gaps, enabling reliable affordance grounding in real-world clutter. Our main contributions are as follows:

- A VLM-guided coarse-to-fine framework that decomposes the challenging problem of language-to-3D affordance grounding into tractable subtasks, integrating 2D priors with VLM reasoning to achieve efficient and precise localization.

- An Affordance Proposal Generation (APG) module that decouples affordance prediction from object identity, producing stable and fine-grained affordance masks that remain robust to viewpoint changes and generalize well across diverse and cluttered scenes.

- An Visual Prompted Affordance Reasoning(VPAR) module fine-tuned via GRPO, that explicitly rewards choices satisfying compositional and relational constraints in a one-of-many setting, thereby selecting the APG-generated mask candidate that best matches the instruction.

## 2 RELATED WORK

**Affordance Grounding**  Affordance grounding aims to identify "action possibilities" of object parts. Early 2D image-based methods range from fully supervised approaches that learn from pixel-level annotations (Zhao et al., 2020; Nguyen et al., 2023; Qian & Fouhey, 2023) to weakly supervised ones that leverage human demonstration videos (Fang et al., 2018; Luo et al., 2024) or learn hotspots on datasets (Hadjivelichkov et al., 2023; Luo et al., 2024; Chen et al., 2024). However, these methods typically require extensive 2D training data and produce view-dependent predictions, and—crucially—are mostly evaluated on clean, object-centric imagery rather than cluttered scenes, limiting their transfer to 3D settings. Subsequent efforts like OOAL (Li et al., 2024a) reduce data dependency via foundation models, yet they still focus on simple verb–noun affordances and struggle with compositional language instructions found in real-world tasks.

To incorporate richer semantic reasoning, recent works integrate Large Language Models (LLMs) with affordance detection. Methods such as LASO (Li et al., 2024c) and 3DAffordanceLLM (Chu et al., 2025) use LLMs to reason about action–object relationships, but are commonly assessed on single, isolated objects under curated conditions, making scaling to cluttered, multi-object scenes challenging. Scaling to complex scenes further requires robust 3D representations. Purely 3D methods—those learning open-vocabulary embeddings for point clouds (Nguyen et al., 2023; 2024; Van Vo et al., 2024) or segmenting functional elements—offer viewpoint invariance. Among them, OpenMask3D-F (Delitzas et al., 2024) and  (Kerr et al., 2023) stands out as an exception that demonstrates affordance prediction in complex, cluttered scenes. Aside from this exception, most existing 3D approaches rely on dense annotations, suffer from data sparsity, and are evaluated on clean, object- or part-centric point clouds, which can lack the semantic depth needed for fine-grained instruction grounding in real-world environments.

**Multimodal Large Language Models (MLLMs)**  MLLMs (Liu et al., 2023; Li et al., 2023) have revolutionized coarse-level scene understanding by combining strong visual perception with semantic reasoning. They have been applied to tasks like visual question answering and image-level referring expression grounding (Zheng et al., 2025a). Their ability to interpret contextual and relational language cues makes them attractive for guiding 3D tasks (Zhang et al., 2024). However, as noted in recent studies (Zhao et al., 2024; Zheng et al., 2025b), their strength in coarse semantics does not directly translate to precise, fine-grained spatial localization required for part-level affordance grounding. They often fail to identify small, critical parts and their predictions can be unstable under varying viewpoints, limiting their utility as a standalone solution for precise 3D alignment.

Our work bridges the gap between these two lines of research. Unlike purely 3D methods, we leverage MLLMs' semantic strength for efficient scene context reasoning. Conversely, we address the fundamental limitations of 2D-driven MLLM approaches by introducing a coarse-to-fine framework that ensures viewpoint robustness and fine-grained geometric precision through affordance-conditioned proposal generation and instruction-guided selection.

## 3 PRELIMINARY

### 3.1 PROBLEM FORMULATION

Let $\mathcal{P} = \{p_i\}_{i=1}^N$ denote a reconstructed 3D point cloud and $\mathcal{I} = \{I_v\}_{v \in \mathcal{V}}$ the associated calibrated frames, where $v$ indexes time. Each frame has a calibrated projection $\Pi_v : \mathcal{P} \to \Omega_v \subset \mathbb{R}^2$. We follow SceneFun3D (Delitzas et al., 2024), which provides $(\mathcal{P}, \mathcal{I})$ with camera poses, a set of natural-language instructions $\mathcal{Q}$ per scene, and 3D affordance masks for a compact function-centric taxonomy

$$\mathcal{A} = \{\texttt{foot\_push}, \texttt{hook\_pull}, \texttt{hook\_turn}, \texttt{key\_press},$$
$$\texttt{pinch\_pull}, \texttt{plug\_in}, \texttt{rotate}, \texttt{tip\_push}, \texttt{unplug}\}. \tag{1}$$

Given an instruction $q \in \mathcal{Q}$ describing an interaction task (possibly with attributes, ordinals, and spatial/part-of relations), the goal is to predict a 3D affordance mask $\mathcal{M}^{3D} \subseteq \mathcal{P}$ that localizes the *interactable* region referred to by $q$.

To obtain per-view supervision from the dataset's 3D masks while avoiding occlusion bleed-through, for each frame $v$, we construct a visibility-aware pixel→point index map $\text{Id}_v : \Omega_v \to \{1, \ldots, N\} \cup \{\varnothing\}$ and derive occlusion-consistent 2D ground-truth masks $Y_v^a \in \{0, 1\}^{\Omega_v}$ for each $a \in \mathcal{A}$ (Sec. 3.2). These quantities provide clean 2D labels for learning and enable efficient 2D→3D lifting during aggregation.

### 3.2 VISIBILITY-AWARE 3D→2D LABEL GENERATION

**Motivation.** Naive projection of 3D masks to images leads to *occlusion bleed-through* (e.g., parts of a cabinet behind a bed appearing in front views). We therefore compute visibility-consistent projections and cache them.

**Visibility-aware Index Map.** For each frame $v$, we compute a pixel→point map $\text{Id}_v : \Omega_v \to \{1, \ldots, N\} \cup \{\varnothing\}$ using a rendering-inspired procedure adapted from Sheng et al. (2024):

1. *Initial projection and depth buffering.* Project all points $p_i \in \mathcal{P}$ to obtain $(u_i, z_i, i)$ tuples.
2. *Local neighborhood search.* For each pixel $u$, gather nearby projected points $\mathcal{N}_u = \{p_i : \|\Pi_v(p_i) - u\| \leq r\}$ (kd-tree), enforcing spatial locality.
3. *Visible surface selection.* Choose the visible candidate by either (i) fitting a local plane to foreground candidates in $\mathcal{N}_u$ and intersecting the camera ray, or (ii) a tolerant z-buffer that selects $\arg\min_{i \in \mathcal{N}_u} z_i$ while merging nearly coplanar hits within depth tolerance $\delta$. Set $\text{Id}_v(u)$ to the chosen point index, or $\varnothing$ if none.

We cache $\text{Id}_v$ for reuse in both training and inference.

**2D Ground-Truth Masks.** A frame can contain several affordance types and several instances of each type. For each affordance $a \in \mathcal{A}$, let the dataset provide 3D instance masks $\mathcal{S}_a = \{\mathcal{M}_{a,k}^{3D} \subseteq \mathcal{P}\}_{k=1}^{K_a}$. For frame $v$, we project each instance to obtain an instance-level mask

$$Y_v^{a,k}(u) = \mathbb{K}\left[\text{Id}_v(u) \in \text{idx}(\mathcal{M}_{a,k}^{3D}) \text{ or } \exists p \in \mathcal{M}_{a,k}^{3D} : \|\Pi_v(p) - u\| \leq r, \ |z(p) - z(\widehat{p}(u))| \leq \delta\right], \tag{2}$$

where $\widehat{p}(u)$ is the visible point at pixel $u$. We then form a per-affordance mask by taking the union over instances:

$$Y_v^a(u) = \mathbb{K}\left[\sum_{k=1}^{K_a} Y_v^{a,k}(u) > 0\right]. \tag{3}$$

Collecting all $a \in \mathcal{A}$ gives a multi-label target $\mathbf{y}_v(u) \in \{0, 1\}^{|\mathcal{A}|}$ with $[\mathbf{y}_v(u)]_a = Y_v^a(u)$.

### 3.3 GROUP RELATIVE POLICY OPTIMIZATION (GRPO)

Group Relative Policy Optimization (GRPO) (Shao et al., 2024) is a reinforcement learning technique that refines a policy $\pi_\theta$ by learning from a group of self-generated responses $\{y_i\}_{i=1}^N$. Each response is assigned a reward $R(y_i)$, and the group's statistics are used to compute a standardized

advantage: $\hat{A}_i = \frac{R(y_i) - \bar{R}}{\text{std}(\{R(y_j)\}_{j=1}^N) + \epsilon}$. where $\bar{R}$ is the mean reward of the group and $\epsilon$ provides numerical stability. The policy is then updated to maximize the following KL-regularized objective:

$$\mathcal{L}_{\text{GRPO}}(\theta) = \mathbb{E}_{y_i \sim \pi_\theta} \left[ \hat{A}_i \log \pi_\theta(y_i) \right] - \beta \, \text{KL} \left( \pi_\theta \| \pi_{\text{ref}} \right). \tag{4}$$

## 4  OUR APPROACH

ThinkAfford follows a multi-stage coarse-to-fine pipeline (Fig. 1). We assume the visibility-aware pixel→point index map $\text{Id}_v$ and per-affordance 2D labels $Y_v^a$ are available from Sec. 3.2. First, the coarse stage performs *Instruction Parsing* followed by a lightweight *View Selection* to keep a compact set of frames $\mathcal{V}^\star$ likely to contain the described context (Sec. 4.1). Then, the fine stage proceeds in two steps: (i) an **Affordance Proposal Generation (APG)** module predicts dense affordance heatmaps and extracts candidate masks via an *Affordance Proposal Network (APN)*; (ii) a **Visual Prompted Affordance Reasoning (VPAR)** uses GRPO to *select* the correct region(s) *within each view* (Sec. 4.2). Finally, during 2D→3D back-projection, we *reuse the same GRPO-trained selector as a strict Y/N ranker* to score the per-view selection and weight multi-view fusion (Sec. 4.3).

### 4.1  COARSE STAGE: INSTRUCTION PARSING AND VIEW SELECTION

**Instruction Parsing.**  A VLM parses $q$ into $(t, \mathcal{A}_{\text{ctx}}, \mathcal{R})$: target $t$, contextual anchors $\mathcal{A}_{\text{ctx}}$, relations $\mathcal{R}$. For instance, given "*Open the fifth drawer of the cabinet to the left of the TV*", VLM extracts: target $t$ as "*drawer*" with ordinal attribute "*fifth*", contextual anchors $\mathcal{A}_{\text{ctx}}$ including "*cabinet*" (primary) and "*TV*" (secondary), and spatial relations $\mathcal{R}$ encoding "*part_of(drawer, cabinet)*" and "*left_of(cabinet, TV)*".

**View Selection.**  We achieve view selection in two steps:

**(a) Grid-based presence probing:** We sample every $\tau$-th frame, assemble thumbnails into mixed $(1{\times}4)/(2{\times}4)$ grids using adaptive layouts, and query a presence probe for the *primary anchor* (fallback to $t$), returning the IDs of frames containing the anchor (if any). For each grid image $G_k$ containing thumbnail frames $\{f_i\}$, we apply VLM-based presence detection: $\mathcal{C}_k = \text{VLM}(G_k, q_{\text{probe}}) \to \{\text{frame\_ids}\}$, where $q_{\text{probe}}$ targets the primary context anchor when $\mathcal{A}_{\text{ctx}} \neq \emptyset$. The union forms $\mathcal{C} = \bigcup_k \mathcal{C}_k$.

**(b) Neighborhood expansion and ranking:** For each $u \in \mathcal{C}$, expand a temporal neighborhood $\mathcal{N}(u) = \{v : |v - u| \leq n, \text{stride} = \Delta\}$ and merge to $\mathcal{V}$. We then query a compact Y/N prompt asking whether $I_v$ matches the text-only description, and compute $s_{\text{coarse}}(v) = P(Y \mid q, I_v)$. Instead of a fixed threshold, we sort $\mathcal{V}$ by $s_{\text{coarse}}(v)$ and keep a Top-$K$ subset with bounds:

$$\mathcal{V}^\star = \text{TopK}_{v \in \mathcal{V}} \big( s_{\text{coarse}}(v), K \big), \qquad K = \min \big( \max(\text{min\_keep}, |\mathcal{V}|), \text{max\_keep} \big).$$

When $|\mathcal{V}| < \text{min\_keep}$, we pad by adding nearest temporal neighbors of the top-ranked frames (without replacement). Unless otherwise stated, we set $\text{min\_keep}{=}2$ and $\text{max\_keep}{=}20$ by default.

### 4.2  FINE STAGE: AFFORDANCE PROPOSAL GENERATION AND VISUAL-PROMPTED AFFORDANCE REASONING

For each $v \in \mathcal{V}^\star$ we localize interactable regions in two steps: (1) the **Affordance Proposal Generation (APG)** module predicts the affordance type and generates affordance-conditioned segmentation masks via an *Affordance Proposal Network (APN)*; (2) the **VAPR** *selects* the region(s) that satisfy the instruction within $I_v$ using GRPO (Sec. 4.2.2). The selected per-view mask is then used for 2D→3D lifting and multi-view fusion (Sec. 4.3).

#### 4.2.1  AFFORDANCE PROPOSAL GENERATION (APG)

**Affordance Label Prediction.**  We first map the instruction $q$ and view $I_v$ to a discrete functional category from the predefined affordance taxonomy $\mathcal{A}$ (Sec. 3.1). This is achieved by prompting a VLM to analyze the image and instruction, canonicalizing its textual output to yield the predicted affordance label $a^\star \in \mathcal{A}$.

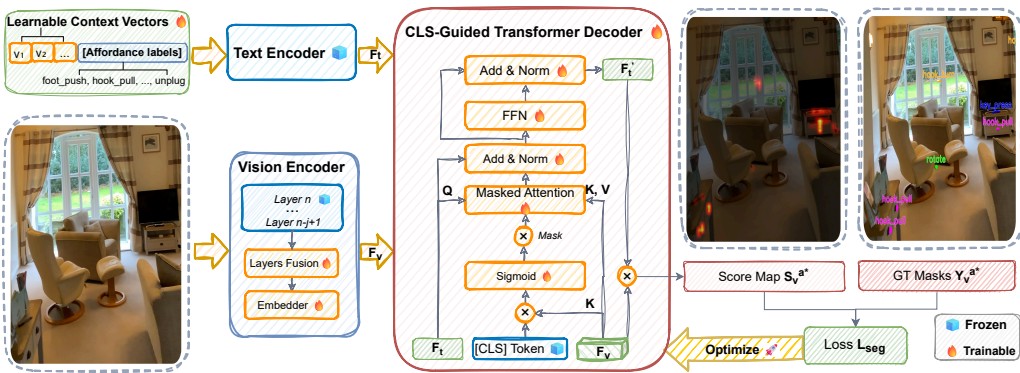

Figure 2: **An illustration of the Affordance Proposal Generation module.** Frozen CLIP/DINOv2 encoders with learnable context vectors feed a CLS-guided decoder to produce affordance heatmaps.

**Affordance Proposal Network (APN).** **(1) Architecture and Training.** We instantiate an *Affordance Proposal Network (APN)* following OOAL (Li et al., 2024a): frozen DINOv2 vision and CLIP text encoders with a lightweight CLS-guided cross-attention decoder, specifically adapted for complex multi-object scenes with occlusions and clutter. We incorporate CoOp (Zhou et al., 2022) for affordance-specific prompt learning and multi-layer feature fusion. During *training*, the APN is conditioned on the ground-truth affordance type $a$ to produce heatmaps $S_v^a$; during *inference*, it is conditioned on the predicted type $a^\star$ to produce $S_v^{a^\star}$. We train on visibility-aware 2D labels from Sec. 3.2 with per-pixel BCE:

$$\mathcal{L}_{\text{seg}} = \text{BCE}(S_v^a, Y_v^a). \tag{5}$$

Backbones are frozen; only context vectors and the decoder are updated.

**(2) Inference.** At inference, we resize $S_v^{a^\star}$ to the image size and threshold at $\gamma$ to obtain a foreground mask. Foreground pixels are clustered with DBSCAN to separate instances; each cluster forms a candidate region. We keep the resulting set $\mathcal{M}_v = \{M_{v,j}\}_{j=1}^{J_v}$, each proposal carries a confidence (mean score inside the mask) and a letter/ID overlay (A, B, ..., Z, AA, ...) for downstream selection.

### 4.2.2 VISUAL-PROMPTED AFFORDANCE REASONING (VPAR)

**Action Space and Grammar.** When sampling rollouts, the selector takes the instruction $q$ and an annotated image $I_v'$ with letter-tagged proposals, generates a natural-language reasoning process, followed by the final choice using the same letter tags and grounded in $/boxed\{\}$ format. We parse the letters into indices $\hat{S} \subseteq \{1, \ldots, J_v\}$ via a base-26 mapping $\phi : \{A, B, \ldots\} \rightarrow \{1, \ldots, J_v\}$ (A$\mapsto$1, B$\mapsto$2, ..., Z$\mapsto$26, AA$\mapsto$27, etc.).

**Optimization with GRPO.** The ground-truth index set $S^\star$ is constructed from projected instance masks when available; 2D IoU$\geq \theta$ links GT instances to proposals, default $\theta=0.5$. We use a structured reward to calculate a comparison between the parsed answer and ground truth:

$$R_{\text{fmt}}(y) = \mathbb{1}[\text{valid boxed output}], \tag{6}$$

$$R_{\text{acc}}(\hat{S}, S^\star) = J(\hat{S}, S^\star) \cdot \left(1 - \tfrac{1}{2}\left(\tfrac{|S^\star \setminus \hat{S}|}{\max(1, |S^\star|)} + \tfrac{|\hat{S} \setminus S^\star|}{\max(1, |\hat{S}|)}\right)\right), \tag{7}$$

$$R(y, S^\star) = (1 - \lambda_{\text{fmt}}) R_{\text{acc}} + \lambda_{\text{fmt}} R_{\text{fmt}}. \tag{8}$$

We then apply the rewards to calculate advantages and update our policy network with the optimization target introduced in ection 3.

**Inference (Per-View Selection).** At inference, for each kept view $v$, we decode the boxed set $\hat{S}$ and obtain the selected indices. We adopt a top-1 policy per view: $j^\star = \arg\max_{j \in \hat{S}} \text{conf}(M_{v,j})$ (or break ties by mask area), yielding the per-view chosen mask $M_{v,j^\star}$.

Figure 3: **An illustration of VAPR module with GRPO.** The policy samples boxed selections; a structured reward (format + penalized Jaccard) drives GRPO updates.

**Reasoning-enhanced Multi-view Fusion.** Based on the fact that the VAPR policy is explicitly rewarded for understanding attribute, relational, and geometric constraints. The model demonstrates favorable ability of binary judgment on a chosen per-view mask naturally reflects instruction fidelity. We therefore reuse it as a strict binary ranker to weight multi-view fusion: we highlight $M_{v,j^\star}$ in $I'_v$ and ask a single-token yes or no question reflecting all constraints in $q$, obtaining

$$r(v, j^\star) = P(Y \mid q, \ I'_v, \ M_{v,j^\star}) \in [0, 1].$$

This score is used as the view weight in Sec. 4.3.

### 4.3 2D→3D Back-Projection

For each kept view $v$ and its chosen mask $M_{v,j^\star}$, we lift pixels to 3D via the cached index map:

$$\mathcal{S}_v = \{ \, \text{Id}_v(u) \ : \ u \in M_{v,j^\star}, \ \text{Id}_v(u) \neq \varnothing \, \} \subseteq \mathcal{P}.$$

We weight each view by the *selector-reused* Y/N confidence $w_v = r(v, j^\star) \in [0, 1]$. For each point $p_i$ we compute the weighted evidence

$$\sigma(p_i) = \frac{\sum_{v \in \mathcal{V}^\star} w_v \, \mathbf{1}[\, p_i \in \mathcal{S}_v \,]}{\sum_{v \in \mathcal{V}^\star} w_v}, \tag{9}$$

and obtain the final 3D affordance mask by thresholding ($\eta$) :

$$\mathcal{M}^{3D} = \{ \, p_i : \sigma(p_i) \geq \eta \, \}.$$

Majority is recovered by $w_v \equiv 1$. If multiple overlays exist at the same timestamp, we keep the mask with the highest $w_v$ to avoid double counting.

## 5 Experiments

### 5.1 Experimental Setup

**Dataset and Evaluation Metrics.** In this work, we conduct experiments on SceneFun3D (Delitzas et al., 2024), currently the only public dataset with annotations for scene-level task-driven 3D affordance grounding. This dataset is officially partitioned into *train* (200 scenes), *val* (30 scenes), and *test* (85 scenes) sets. Each indoor scene contains high-resolution RGBD scans with calibrated camera poses (on average ∼1800 images per scene). Per scene, the dataset provides ∼15 natural-language task descriptions paired with 3D affordance masks; a single instruction may correspond to multiple instances, and the ground-truth mask includes all valid instances. Following the standard evaluation protocol of SceneFun3D, we measure performance using Average Precision (AP) at IoU thresholds of 0.25 and 0.5 (denoted as AP25 and AP50). We primarily use the *val* set for ablation studies and compare with other methods on both the *val* set and the *test* set. The performance on the *test* set is obtained through submitting the results to the official test server.

Table 1: Performance comparison with state-of-the-art methods on SceneFun3D. We highlight the best and second-best performance with **bold font** and underline.

| Method | Test Set | | Val Set | |
|---|---|---|---|---|
| | AP25 | AP50 | AP25 | AP50 |
| *W/o VLM:* | | | | |
| OpenMask3D Takmaz et al. (2023) | - | - | 0.00 | 0.00 |
| LERF Kerr et al. (2023) | - | - | 11.30 | 4.90 |
| OpenMask3D-F Delitzas et al. (2024) | - | - | 17.50 | 8.00 |
| *VLM-based:* | | | | |
| Fun3DU Corsetti et al. (2025) | 8.80 | 3.56 | 17.33 | 6.41 |
| ThinkAfford (Ours) | **14.97** | **4.44** | **22.69** | **8.60** |

**Implementation Details.** We adopt Qwen2.5-VL-7B Bai et al. (2025) as the base VLM in our pipeline. The coarse stage is performed in a zero-shot manner, without finetuning the VLM. In the fine stage, APN and VPAR are trained separately. For APN, we use frozen DINOv2 for vision and CLIP for text encoders, coupled with a lightweight CLS-guided cross-attention decoder. The model is optimized using SGD with a learning rate of 0.01, trained for 200000 steps with a batch size of 1. For VPAR training, we fine-tune the VLM using GRPO for 15 epochs with a learning rate of 1e-6, AdamW optimizer, and batch size of 128. During rollout, we generate 8 responses per prompt with temperature 1.0 and top-p 0.99. The KL coefficient is set to 0.01 to control policy deviation.

## 5.2 MAIN RESULTS

We compare ThinkAfford against 3D open-vocabulary baselines (LERF Kerr et al. (2023), Open-Mask3D Takmaz et al. (2023), and OpenMask3D-F Delitzas et al. (2024)) and the VLM-based method Fun3DU Corsetti et al. (2025). As shown in Table 1, ThinkAfford achieves the best performance on both validation and test splits. On the test set, ThinkAfford surpasses Fun3DU by 6.17% AP25, marking a relative improvement of 70.1%. On the validation set, it reaches 22.69% AP25 and 8.60% AP50, outperforming the strongest competitor OpenMask3D-F by 7.5% and 29.7%, respectively. These gains highlight the effectiveness of our coarse-to-fine decomposition. The pronounced improvement in AP50 indicates that ThinkAfford not only identifies more correct regions but also provides finer spatial localization. Performance improvements are consistent across validation and test sets, demonstrating strong generalization under diverse scene layouts and instructions. Compared with OpenMask3D-F, which lacks robust language–geometry alignment, our method benefits from VLM-guided relational reasoning in the coarse stage and targeted fine-grained selection in the fine stage. Furthermore, the clear advantage over Fun3DU, despite its strong 2D segmentation base model, underscores the importance of integrating depth and multi-view cues for precise 3D affordance grounding.

## 5.3 ABLATION STUDY

We conduct several groups of ablative experiments to experimentally study the effect of our design in the proposed ThinkAfford framework. The models are all evaluated on SceneFun3D *val* set.

**Effect of Different Proposed Modules.** Table 2 quantifies the contribution of each component. Removing Grid-based presence probing (GPP) leads to an AP25/AP50 drop of 2.94%/0.42%, indicating that early pruning of off-topic frames reduces search noise and benefits recall. Disabling Neighborhood expansion and ranking causes a 6.82%/1.62% decrease, showing that contextual, relational scoring is critical for selecting instruction-relevant views. Replacing ranker-weighted fusion with uniform averaging degrades multi-view consistency by 3.25%/0.01%, confirming that weighting by ranking confidence mitigates cross-view contamination. Finally, omitting GRPO fine-tuning of the selector (VAPR) yields the largest AP25/AP50 drop (7.28%/2.75%), highlighting its role in enforcing attribute and relational fidelity; AP50 is especially sensitive to fine localization. Overall, the full system improves both recall and precision, with GRPO and ranker-weighted fusion contributing most to spatial accuracy, and SPP/ranking contributing most to robustness in clutter.

Table 2: Ablative experiments on different proposed modules in our ThinkAfford .

| Method | Val | |
|---|---|---|
| | AP25 | AP50 |
| ThinkAfford | **22.69** | **8.60** |
| w/o Grid-based presence probing | 19.75 | 8.18 |
| w/o Neighborhood expansion and ranking | 15.87 | 6.98 |
| w/o Reasoning-enhanced multi-view fusion | 19.44 | 8.59 |
| w/o GRPO RFT | 15.41 | 5.85 |

Table 3: Effect of the number of selected frames in the coarse stage.

| # Selected Frames | Val | |
|---|---|---|
| | AP25 | AP50 |
| 5 | 7.34 | 3.64 |
| 10 | 12.10 | 4.77 |
| 15 | 14.13 | 5.32 |
| 20 | **22.69** | **8.60** |
| 30 | 20.23 | 6.37 |
| 40 | 19.24 | 8.27 |

Table 4: Effect of the voting threshold in 2D-to-3D lifting.

| Threshold | Val | |
|---|---|---|
| | AP25 | AP50 |
| 0.1 | 16.96 | 2.65 |
| 0.2 | 18.95 | 5.14 |
| 0.3 | **22.69** | **8.60** |
| 0.4 | 19.80 | 7.16 |
| 0.5 | 15.32 | 5.28 |
| 0.6 | 9.64 | 3.92 |

**Effect of the Number of Selected Frames.** Table 3 illustrates the effect of varying the number of selected frames. When the view budget is small (fewer than 15 frames), both AP25 and AP50 remain low because the limited perspectives fail to capture sufficient instruction-relevant evidence. As the budget increases, performance improves steadily and reaches its peak at 20 frames (AP25 = 22.69%, AP50 = 8.60%), which strikes a good balance between broad coverage and multi-view consistency. Beyond this point (30/40 frames), performance begins to decline, suggesting that an excessive number of views introduces visual clutter and cross-view inconsistencies that hinder fine localization. We therefore adopt 20 selected frames as the default setting in all experiments.

**Effect of the Voting Threshold in 2D-to-3D Lifting.** Table 4 shows that raising the threshold $\eta$ from 0.1 to 0.3 effectively suppresses cross-view noise, leading to notable improvements in AP50 (+5.95% and +5.37%, respectively). However, setting the threshold too high can be detrimental: at $\eta$=0.6, many partially occluded yet correct proposals are discarded, causing AP25 to drop by 13.05% compared with $\eta$=0.3. Overall, $\eta$=0.3 yields the best validation performance (AP25=22.69%, AP50=8.60%), suggesting that enforcing stronger multi-view agreement is particularly beneficial for precise localization.

## 6 CONCLUSION

This work introduces ThinkAfford, a two-stage RGB-D framework for scene-level 3D affordance grounding. By decomposing the task into VLM-guided view selection and affordance-conditioned proposal selection with VLM reasoning further enhanced by reinforcement learning, ThinkAfford effectively bridges the gap between language instructions and fine-grained 3D affordance localization. The Affordance Proposal Generation (APG) provides robust affordance masks across cluttered scenes and varying viewpoints, while the Visual Prompted Affordance Reasoning (VAPR), optimized with GRPO, enhances the VLM's ability to reason about small objects, compositional relations, and spatial constraints. Extensive experiments validate the superiority of our framework, showing consistent improvements over prior approaches in both granularity and reliability. We believe ThinkAfford establishes a strong framework for advancing instruction-driven affordance grounding and offers significant potential for embodied AI systems operating in complex real-world environments, which will inspire future investigation.

## LLM Usage Statement

Large language models (LLMs) were used solely as an assistive tool for *language polishing* and *typo/wording error checking* throughout the manuscript. No ideas, methods, code, results, or analyses were generated by LLMs. All content was verified by the authors.

## Reproducibility Statement

All experiments are conducted on publicly available datasets, with dataset descriptions and preprocessing steps in Section 5.1. Implementation details of ACP and GRS, training procedures, and hyperparameters are given in Section 5.1; further qualitative results appear in the appendix. The codebase will be made public upon acceptance.

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
