# OpenReview forum: "ThinkAfford: Enhancing VLM Reasoning for Affordance Grounding in 3D Scenes"
_ICLR.cc/2026/Conference — ICLR 2026 Conference Withdrawn Submission_

### Official Review · Reviewer_owXp · 2025-10-28

**Soundness:** 3
**Presentation:** 3
**Contribution:** 3
**Rating:** 4
**Confidence:** 3

**Summary:**

The paper proposes a framework ThinkAfford that infers object affordances in complex 3D environments. ThinkAfford is corse-to-fine framework. In the coarse stage, a vision-language model is used to efficiently reduce thousands of frames to a compact set of relevant candidate views by leveraging contextual and relational cues. In the fine stage, the model focuses on functional parts by generating affordance-centric proposals that stay consistent across viewpoints. An instruction-guided selector, fine-tuned with Group Relative Policy Optimization (GRPO), enhances fine-grained spatial reasoning by explicitly rewarding selections that meet attribute, relational, and geometric constraints. Experiments on SceneFun3D show promising results over the baselines.

**Strengths:**

1. Lifting 2D masks to 3D affordance is interesting, where 2D affordance can be obtained from pretrained VLMs.
2. Overall, this paper is well organized though some small parts seem unclear.
3. Experimental results show that ThinkAfford outperforms the baselines significantly in 3D affordance detection tasks.

**Weaknesses:**

1. The assumption that the visibility-aware pixel→point index map and per-affordance 2D labels are available may not hold in real-world, which hinders the applicability of the proposed framework.
2. The two-stage pipeline, involving view pruning and affordance proposal generation, introduces additional complexity.
3. The reliance on RGB-D data restricts the model applicability in real-world scenarios where the depth data are not available.
4. The final performance of 3D strongly relies on 2D mask generation on the first stage. When the candidates are not good, the final performance cannot be promised.

**Questions:**

1.  The final 3D performance appears to heavily rely on the quality of 2D mask generation. Could the authors clarify how accurate and robust the mask candidates are in the first (corse) stage?
2. It would be helpful to see ablation studies using different vision-language models in the coarse stage, to better assess the contribution and importance of the first stage.
3. How does the computational overhead of the proposed two-stage framework compare with one-stage methods?
4. The comparison in Table 1 looks limited. Could the authors provide more comprehensive comparisons with recent affordance grounding approaches?

---

### Official Review · Reviewer_Zi8Z · 2025-10-30

**Soundness:** 2
**Presentation:** 3
**Contribution:** 3
**Rating:** 4
**Confidence:** 2

**Summary:**

The paper proposes a coarse-to-fine framework for grounding natural language instructions to affordances in 3D scenes. The framework first prompts VLM to select relevant views from the original large number of frames. It then generates affordance proposals using a light-weight APG module. Afterwards, it uses a reasoning VLM fine-tuned with GRPO to select the correct affordance for each view. Finally, the same reasoning VLM is used to fuse multiple views. The framework is experimented on SceneFun3D dataset and surpasses the performance of baselines.

**Strengths:**

1. The integration of a reasoning VLM for grounding compositional language instructions to fine-grained affordances is an interesting idea, and empirically contributes a lot to the performance.

2. The proposed fine-grained pipeline achieves higher AP25 scores than baselines with a clear margin.

**Weaknesses:**

1. The modular pipeline is a little complicated with many non-trainable hyperparameters. The modular design would accumulate errors from the prior modules to later ones, and cannot be easily corrected from the final results. From ablation studies, we can see that the overall performance is quite sensitive to some hyperparameters, such as voting threshold.

2. The experiments are conducted on only one dataset, which is not thorough enough. It remains unclear whether and to what extent the proposed framework can generalize to other scenes. I think demonstrating the application of the proposed framework to some downstream embodied tasks would largely strengthen the work.

**Questions:**

What is the inference time of the proposed framework? Can you discuss how to apply this method to practical scenarios?

---

### Official Review · Reviewer_29uV · 2025-10-31

**Soundness:** 3
**Presentation:** 3
**Contribution:** 1
**Rating:** 2
**Confidence:** 5

**Summary:**

The paper presents ThinkAfford, a two-stage framework designed to enhance 3D affordance grounding using Vision-Language Models (VLMs) and RGB-D observations. The approach aims to identify actionable object parts in complex 3D environments based on natural language instructions (e.g., “turn on the lamp switch”).

ThinkAfford consists of:
1/ A coarse stage that selects contextually relevant views from multi-view RGB-D data using a VLM-driven relevance scoring mechanism;
2/ A fine stage where an Affordance Proposal Generator (APG) proposes candidate regions, followed by a Visual-Prompted Affordance Reasoning (VPAR) module that leverages reinforcement learning (Group Relative Policy Optimization, GRPO) to select the most semantically consistent region;
3/ A 3D reconstruction step that fuses multi-view 2D predictions into a unified 3D affordance mask.

Experiments on SceneFun3D demonstrate improved AP25 and AP50 scores compared to prior works (OpenMask3D-F, Fun3DU), suggesting better alignment between linguistic cues and 3D visual reasoning.

**Strengths:**

1/ The paper is clearly written, well structured, and the proposed pipeline is logically presented with informative figures. The motivation to integrate VLM reasoning with 3D affordance grounding is sound and relevant to embodied AI.
2/ The use of GRPO for optimizing affordance selection through reward signals is an interesting direction, aligning with recent attempts to train VLMs beyond static instruction tuning.

**Weaknesses:**

1/ While the framework performs well, it largely repackages existing ideas — multi-view selection, mask proposal, and VLM-based scoring — into a sequential pipeline. Each module (e.g., DINOv2 feature extraction, CLIP-based region-text matching, GRPO fine-tuning) relies on well-established techniques. The overall contribution is incremental, not conceptually new.
2/ The approach primarily projects 2D affordance masks back into 3D rather than reasoning directly within the 3D space. This weakens the claim of “3D affordance reasoning,” as the model’s spatial understanding still stems from 2D feature correlations.
3/ Much of the performance gain seems to come from the strong underlying VLM (Qwen2.5-VL-7B). The paper does not convincingly show what ThinkAfford adds beyond leveraging existing large models.
4/ All experiments are confined to a synthetic dataset. There is no demonstration of real-world generalization or deployment in physical scenes, making it unclear whether the system can handle domain shifts or noisy sensory inputs.
5/ The reward design in GRPO and the role of the “visual prompts” in VPAR are insufficiently studied.

**Questions:**

Please see my weakness section.

---

### Note · Authors · 2025-11-14

I have read and agree with the venue's withdrawal policy on behalf of myself and my co-authors.